# SARS-CoV-2 Infection in Late Pregnancy and Childbirth from the Perspective of Perinatal Pathology

**DOI:** 10.3390/jdb11040042

**Published:** 2023-11-16

**Authors:** Larisa Debelenko

**Affiliations:** Department of Pathology and Cell Biology, Columbia University—Irving Medical Center, New York, NY 10032, USA; ld2863@cumc.columbia.edu

**Keywords:** SARS-CoV-2, placentitis, stillbirth

## Abstract

This review focuses on SARS-CoV-2 infection in placental and fetal tissues. Viremia is rare in infected pregnant women, and the virus is seldom amplified from placental tissues. Definite and probable placental infection requires the demonstration of viral RNA or proteins using in situ hybridization (ISH) and immunohistochemistry (IHC). Small subsets (1.0–7.9%, median 2.8%) of placentas of SARS-CoV-2-positive women showed definite infection accompanied by a characteristic histopathology named SARS-CoV-2 placentitis (SP). The conventionally accepted histopathological criteria for SP include the triad of intervillositis, perivillous fibrin deposition, and trophoblast necrosis. SP was shown to be independent of the clinical severity of the infection, but associated with stillbirth in cases where destructive lesions affecting more than 75% of the placental tissue resulted in placental insufficiency and severe fetal hypoxic–ischemic injury. An association between maternal thrombophilia and SP was shown in a subset of cases, suggesting a synergy of the infection and deficient coagulation cascade as one of the mechanisms of the pathologic accumulation of fibrin in affected placentas. The virus was amplified from fetal tissues in approximately 40% of SP cases, but definite fetal involvement demonstrated using ISH or IHC is exceptionally rare. The placental pathology in SARS-CoV-2-positive women also includes chronic lesions associated with placental malperfusion in the absence of definite or probable placental infection. The direct viral causation of the vascular malperfusion of the placenta in COVID-19 is debatable, and common predispositions (hypertension, diabetes, and obesity) may play a role.

## 1. Introduction

Since the beginning of the COVID-19 pandemic, complications in pregnancy and adverse clinical outcomes in the mother and offspring have been the focus of attention. Several studies using large cohorts of pregnant women showed that those with COVID-19 infection were more likely to have preterm and cesarean delivery; require intensive care, intubation, and mechanical ventilation; and have fatal hospital outcomes than uninfected pregnant women [1,2]. Data from the National Database in England and the US Center for Disease Control and Prevention (CDC) demonstrated that pregnant women with COVID-19 had increased risks of stillbirth compared to their uninfected counterparts [3,4]. Placental examination and fetal and neonatal autopsies provide unique opportunities to study the mechanisms that explain maternal and fetal morbidity and fetal demise. This review focuses on recently discovered disease processes associated with or caused by the infection and detected in placental and fetal tissues.

## 2. SARS-CoV-2 Infection and Chronic Placental Pathology

The current practice of placental pathology is based on the international consensus classification, the so-called Amsterdam criteria [5], which include acute amniotic fluid infection and chronic placental lesions (maternal vascular malperfusion (MVM), fetal vascular malperfusion (FVM), and inflammatory lesions featuring chronic villitis, chronic lymphoplasmacytic deciduitis, and chronic histiocytic intervillositis (CHI)).

One of the first reports on placental pathology in women with moderate to severe symptomatic COVID-19 showed features of MVM, namely, decidual arteriopathy (DA), including mural hypertrophy, atherosis, and the fibrinoid necrosis of decidual arteries [6]. Interestingly, in this report, DA, which is generally considered the hallmark of preeclampsia (PE), was not associated with maternal hypertension in infected women. DA is thought to be initiated in the second trimester by inadequate placental implantation [7], and the development of this pathology in the third trimester of SARS-CoV-2 infection remains unexplained.

Another study on the placentas of SARS-CoV-2-positive women published in the first months of the pandemic showed an increased tendency for FVM, explained by the hypercoagulation associated with the infection [8].

The findings of fetal and maternal vascular malperfusion in the placentas of SARS-CoV-2-positive women were confirmed by later studies [9,10,11]; however, a number of reports did not support the etiologic role of SARS-CoV-2 in placental malperfusion since the virus was not identified in placental tissues, and previously established antenatal risk factors (PE, diabetes, obesity, and chronic respiratory and cardiovascular disease) were involved [12,13,14,15,16,17].

A recent study on placental pathology noted maternal vascular malperfusion in 16 of 44 (36%) SARS-CoV-2-exposed placentas and 8 of 44 (18%) unexposed placentas (*p* = 0.06) with the odds of maternal vascular malperfusion lesions increasing significantly with disease severity [11]. In all of these cases, no virus was amplified from the placental tissues.

MVM is known to be associated with systemic maternal disorders, particularly hypertensive disorders (PE, pregnancy-induced hypertension, and HELLP) [5]. PE is a serious complication of pregnancy that is diagnosed when gestational hypertension is accompanied by new-onset proteinuria. Reduced placental perfusion is considered the “root cause” of PE, which may translate into the multisystemic maternal syndrome, characterized by vasoconstriction, metabolic changes, endothelial dysfunction, the activation of the coagulation cascade, and an increased inflammatory response [18]. Inadequate placentation and syncytiotrophoblast stress lead to the increased expression of soluble fms-like tyrosine kinase-1 (sFlt-1) and reduced placenta-derived growth factor (PlGF), causing endothelial damage and dysfunction, which plays a central role in this disorder. PE is underlined by a number of genetic, behavioral, and environmental factors, which are modified by the normal physiological changes of pregnancy with the particular relevance of an increased inflammatory response. Conditions known to increase the risk of preeclampsia include obesity, hypertension, diabetes, hyperhomocysteinemia, increased androgens, and Black race [19].

Although early reports indicated increased rates of hypertension in COVID-19-infected pregnant women [20], several population-based and longitudinal studies did not confirm an association between PE and COVID-19 [21,22]. COVID-19 during pregnancy may manifest as a PE-like syndrome, which is differentiated from true PE by lower sFlt-1-to-PlGF ratios [23]. Unlike PE, PE-like syndrome disappears when symptoms of acute infection subside and pregnancy continues. We did not find any literature on placental pathology in PE-like syndrome; it would be interesting to see whether MVM in SARS-CoV-2 is associated with this syndrome.

Redline et al. [24] showed a significantly increased rate of high-grade MVM in preterm placentas and an increased rate of high-grade chronic villitis (negative for the virus using IHC) in the term placentas of women with COVID-19, but favored pre-pregnancy risk factors or COVID-19-related increased susceptibility, rather than direct viral causation, as a potential explanation of these findings.

Thus, the question of the etiological role of SARS-CoV-2 in chronic placental pathology (MVM, FVM, chronic villitis) observed in COVID-19 pregnancies remains open, and more recent analyses suggest that this pathology can be explained by previously known risk factors (obesity, gestational diabetes, hypercoagulation states, immunological dysregulation) responsible for susceptibility to chronic placental lesions, as well as COVID-19 [24].

## 3. Identification of SARS-CoV-2 in Placental and Fetal Tissues

Soon after the beginning of the epidemic, several studies reported the amplification of viral RNA from swabs and homogenates of placental tissues [25,26]. These SARS-CoV-2 PCR-positive placentas were not associated with the transmission of the virus to fetuses or neonates and/or placental pathology, raising the question of maternal viremia as a source of the positive PCR results in this highly vascularized organ with a significant input of circulating maternal blood.

Starting in May 2020, several case reports demonstrated the virus in placental tissues by electron microscopy, ISH and IHC, in association with trans placental infection and placental inflammation localized to the intervillous spaces [16,27,28,29,30,31,32]. These cases also showed that villous trophoblasts, the cellular layer of the placental barrier, were the main targets of SARS-CoV-2 and exhibited strong ISH and IHC signals reflecting the accumulation of viral RNA and proteins in these cells (Figure 1).

In August 2021, the NIH expert consensus published standardized criteria for definite, probable, possible, and unlikely SARS-CoV-2 infection of the placenta [33]. Definitive infection required the demonstration of the replicating virus in the placental tissue using ISH to either antisense strand or double-stranded RNA. Infection was deemed probable if the virus could be detected in the placental tissue via ISH with the sense RNA probe or IHC. Infection was considered possible if it was detected through PCR of washed placental tissue homogenates or electron microscopy. If all of the above tests were negative, placental infection was considered unlikely. These criteria did not take into account placental inflammation (placentitis), which was later found to be a characteristic manifestation of SARS-CoV-2 infection identified in a subset of pregnancies.

## 4. SARS-CoV-2 Placentitis

The first reports of SARS-CoV-2 placental infection proven using ISH and/or IHC described the associated pathology as CHI [16,22,23,24,25,26,27,28,29,30,31], a predominantly histiocytic inflammation localized in the maternal compartment of the placenta (intervillous spaces). This pattern of placental inflammation, in severe cases accompanied by trophoblast necrosis and perivillous fibrin deposition (PVFD), was classically described by Lebare and Mullen in 1987 without a clear association with infections [34].

To describe the SARS-CoV-2 placental inflammation, Linehan et al. [32] used the term “placentitis”, and later, Watkins et al. [35] analyzed seven positive cases and defined the term “SARS-CoV-2 placentitis” (SP) as a histopathological triad that included histiocytic and neutrophilic intervillositis, PVFD, and trophoblast necrosis. The subsequent reports confirmed that the histopathological triad of SP was robust and reliable, facilitating the recognition of new cases of SARS-CoV-2 infection in the placenta. The term SP has become universally accepted [24,36,37,38,39,40,41].

SP differs from classical CHI, also known as a “massive histiocytic intervillositis” and characterized by the diffuse infiltration of intervillous spaces by histiocytes. The inflammatory infiltrate in SP is patchy and rather scarce and, besides histiocytes, contains various numbers of neutrophils and lymphocytes [16,24,35]. In addition, unlike in CHI, PVFD in SP is marked and frequently massive, occupying more than 25% of the placental disc (and greater than 75% in cases of intrauterine fetal demise [38,41]). Finally, SP is characterized by extremely pronounced trophoblast necrosis due to combined effects of the complement-mediated immunological damage of this cellular layer, direct viral cytopathic change, and associated apoptosis [16,24,35,42].

Based on the morphologic grade and placental distribution, SP was classified into diffuse and localized, and the latter was thought to represent either the early or late (recovering) stages of placental infection [24].

While the histological features of SP are consistent and well defined, the clinical expression of the disease is not uniform, and there is no correlation between this placental pathology, antenatal comorbidities, and the clinical severity of COVID-19. In many SP cases, women were healthy and asymptomatic [16,38,43,44], while the placentas of pregnant patients with moderate to severe COVID-19 showed SP in small subsets of cases [24,38,45].

The prevalence of SP in COVID-19 is difficult to calculate, since many placentas of infected but asymptomatic women remain unexamined on the microscopic level. Most published series represent only positive cases referred from different primary institutions [36,37,39,42,43], making it difficult to determine the prevalence of SP in the population. However, some reports from large institutions, prospective studies, and country registries estimated that SP is rare overall, affecting from 1 to 7.9% (median reported 2.8%) of SARS-CoV-2-positive women at delivery [24,38,46,47].

SP was first reported in association with wild-type SARS-CoV-2 [28] and continued to be reported in subsequent years, when new variants emerged [38,40,46,47]. Several authors have noted a “surge” in positive cases with the delta variant [39,41,47]; however, this may have been related to the improved diagnostics of SP by the time this variant prevailed.

In summary, SP has been established as a morphologically recognizable diagnostic pattern of SARS-CoV-2 infection that affects both maternal (intervillous spaces) and fetal (trophoblast) placental compartments and signifies transplacental viral transmission [48]. The morphological pattern of SP is characteristic, but not pathognomonic, and requires confirmation of SARS-CoV-2 etiology using ISH and/or IHC [35].

Although SP is rare and frequently asymptomatic, it progresses to massive PVFD in a subset of cases, leading to placental insufficiency and fetal demise [36,38,43,49].

## 5. Massive Perivillous Fibrin Deposition as an End-Stage SARS-CoV-2 Placentitis Causing Hypoxic–Ischemic Fetal Death

The first case reports of placental involvement by SARS-CoV-2 were published in mid-2020–early 2021, and in April 2021, the temporal cluster of six stillbirths in women with COVID-19 in Ireland posed a question of possible significant risks of the infection for adverse pregnancy outcomes [50]. Placental examination in these cases revealed SP with massive PVFD, involving more than 80% of the placental discs [36]. The women had either no (*n* = 1) or mild (*n* = 5) symptoms of COVID-19, and three of the six had thrombocytopenia. Decreased fetal movements were documented in three of six cases occurring from 8 to 18 days prior to fetal demise. Postmortem examinations were negative for the involvement of fetal organs and tissues with SARS-CoV-2 in five cases that underwent autopsies.

The most representative analysis of 68 cases of stillbirth and neonatal death collected from mothers whose placentas were diagnosed with SP in 12 countries during 2020–2021 showed that destructive SP morphology (trophoblast necrosis, massive PVFD) involved, on average, more than 77% of placental tissue. In 30 cases, the data from postmortem fetal examinations were available and showed no recurrent anomalies or inflammation. In 19 of 30 cases, there were features of fetal hypoxia and/or asphyxia; in 16 of 30 cases, the SARS-CoV-2 was amplified from fetal nasopharyngeal swabs, and in 4 of these 16 cases, from homogenates of fetal organs as well [43].

A prospective study from Greece presented six cases of stillbirth collected during April 2020–August 2021 from women with the clinical diagnosis of COVID-19 and histopathological diagnosis of SP. In these cases, perivillous fibrin deposition involved > 75% of placental tissues. Two of the six women were asymptomatic, and four had mild COVID-19. Postmortem examinations were performed in five cases and showed features of hypoxic–ischemic injury and negative SARS-CoV-2 PCR results in the fetal tissue homogenates in each case. In seven additional cases of SP that resulted in unaffected live-born neonates, the placental involvement by destructive SP lesions was less than 25% [38].

A retrospective study from Sweden [51] reported five cases of stillbirth and two early neonatal deaths from 13 women with SP. The time between the diagnosis of COVID-19 and fetal demise was 2–25 days; 77% of women had reduced fetal movements. All placentas showed borderline to massive PVFD. The virus was amplified from fetal tissues in two of the five stillborn fetuses.

Cumulatively, these (Table 1) and several other reports [40,41,42,44] demonstrate that SP is a severe pathology associated with fetal mortality in approximately 50% of the reported cases. Common features included diffuse damage of the placental barrier due to the necrosis of villous trophoblasts and the encasement of chorionic villi by fibrin (Figure 2). The process appeared to exceed the placental reserve at approximately 75% involvement when the developed placental insufficiency resulted in fetal asphyxia or hypoxic–ischemic injury that was severe enough to cause fetal death. Interestingly, the fetal demise due to an advanced SP was seen in asymptomatic or mildly symptomatic COVID-19-infected women, and in a substantial proportion of cases, decreased fetal movements, reflecting placental insufficiency, had been reported and lasted 6 days on average before the demise [51]. This brief interval indicates a high velocity of SP progression to massive PVFD.

Massive PVFD, also known as maternal floor infarction, was recognized before the epidemic as a cause of placental insufficiency associated with fetal growth restriction and intrauterine fetal demise, including recurrent pregnancy loss [52]. Its association with chronic placental lesions (chronic villitis and chronic intervillositis) is known; however, its infectious etiologies have not been clearly demonstrated prior to the COVID-19 epidemic. Due to the extended placental involvement and association with fetal growth restriction, massive PVFD was thought to represent a chronic lesion evolving for some time before reaching the end stage seen in the placentas of stillborn fetuses [49]. The new evidence derived from the data on SP proved that infection was one of the etiologies of massive PVFD and demonstrated the rapid progression of this process in SARS-CoV-2 infection without preexisting fetal growth restriction.

Approximately half of the SP cases resulted in live births, and in these cases, PVFD involved less than 75% of the placental tissues. It is logical to assume that, in these cases, the progression of SP to massive PVFD was effectively terminated by labor, preventing placenta-induced hypoxic–ischemic injury to the offspring. However, it appears that not all cases would progress to massive PVFD as localized SP has been described, indicating that the disease may be self-limited when small foci of intervillositis are sequestered within the placenta by the surrounding fibrin [24].

Although SARS-CoV-2 was amplified either from fetal nasopharynx or tissue homogenates cumulatively in approximately 40% of tested and reported stillborn fetuses with SP, the autopsies did not provide morphological or IHC evidence of COVID-19 in the offspring, with rare exceptions [53].

## 6. Statistics of Increased COVID-19-Related Adverse Pregnancy Outcomes

Population-based statistical data indicate that COVID-19 documented at delivery is associated with an increased risk of stillbirth, with a stronger association during the period of delta variant predominance in the United States [4]. Thus, the rates of stillbirth in women without COVID-19 at delivery in this analysis (0.64% overall) were similar to the known pre-pandemic stillbirth rate of 0.59%; however, 0.98% of COVID-19-affected deliveries pre-delta and 2.70% during the delta period resulted in stillbirth. Likewise, a nationwide (Belgium) prospective study indicated the almost twofold increase in the stillbirth rates during the first and second waves of epidemics in Europe [54]. These statistics have not been correlated with placental pathologies, and the proportion of excessive stillbirths due to SP remains undetermined.

## 7. Biology of Placental Response to SARS-CoV-2

Angiotensin-converting enzyme 2 (ACE2) and transmembrane serine protease 2 (TMPRSS2) have been identified as entry factors for SARS-CoV-2 infection [55]. Several studies showed the expression of these receptors in human placental tissue, including the trophoblast [56,57]; however, the levels of this expression were highest early in pregnancy and diminished as pregnancy progressed [11,57], suggesting that low ACE2 expression levels in the third trimester play a protective role against SARS-CoV-2. This may explain the overall low prevalence of SP.

A retrospective cohort study using multilevel logistic regression analyses of nationwide electronic health records in the United States showed that increased risks of stillbirth were only associated with COVID-19 during early and mid pregnancy and not at any time before delivery or during the third trimester [58]. This may also be explained by the lower probability of placental infection due to the decreased expression of SARS-CoV-2 cellular entry molecules in the third trimester. Syncytiotrophoblasts have been shown to be infected through ACE2 and TMPRSS2 in an in vitro model of the early placenta [59]; however, we did not find publications demonstrating the co-expression of the virus and its entry molecules in SP in the third trimester.

Many studies have examined placental inflammation in COVID-19 at the molecular level and found that the maternal–fetal interface of SARS-CoV-2-infected women exhibited robust immune responses, including the increased activation of natural killer (NK) and T cells, increased expression of interferon-related genes, and markers associated with preeclampsia, even in the absence of detectable local viral invasion [60]. The placental expression of interferon-induced transmembrane proteins IFITM1 and IFITM3 implicated in the antiviral response was also increased in severe COVID-19 without SP [48].

In SP, viral proteins have been shown to be co-expressed with the complement marker C4d [24,35], suggesting a role of the complement fixation along the villous borders in the tissue destruction. However, this process is not specific and has been demonstrated in the CHI of noninfectious etiologies [61]. Another finding that may shed light on the pathogenesis of SP is the decreased expression of anti-apoptotic marker BCL-2, implicating apoptosis as a mechanism at least partially responsible for extensive trophoblast damage in SP [24]. The authors proposed the term “aponecrosis” to describe the hybrid lesion of necrosis and apoptosis in the villous trophoblasts in SP cases.

Abnormal coagulation and impaired fibrinolysis due to polymorphisms in the Plasminogen Activator Inhibitor 1 (PAI-1) and 5-methyltetrahydrofolate (MTHFR) genes showed a significant association with SP [38]. Thrombocytopenia was also reported in three of six cases of stillbirth due to SP with massive PVFD by Fitzgerald et al. [36]. These data support the role of the abnormal activation of the maternal coagulation cascade acting in synergy with the infection, leading to the excessive accumulation of fibrinoid extracellular matrix in SP, as proposed by Redline [62].

It has been shown that SARS-CoV-2 viremia is extremely rare in pregnant women regardless of the clinical severity of the infection [11]. Interestingly, the viral RNA was demonstrated in the blood of two of six studied cases of SP, as opposed to 0/12 controls [63]. It is likely that SP arises in rare cases of viremia in pregnancy; however, the underlying factors that predispose placentas to SP are not understood, and the dichotomy of COVID-19 infection with the vast majority of placentas not affected by the virus and a small subset of SP progressing to massive PVFD remains obscure.

Interestingly, almost in all reported SP cases, the women were not vaccinated against SARS-CoV-2 [49]. One case of stillbirth due to SP with massive PVFD was reported in a not-fully-vaccinated parturient who received only one dose of the Pfizer–BioNTech COVID-19 vaccine 25 days prior to testing positive and 4 weeks prior to delivery [39]. Both Pfizer–BioNTech and Moderna SARS-CoV-2 RNA vaccines have been shown to be safe and effective in the production of maternal antibodies that are not only detectable in maternal sera at delivery and in breast milk, but are also present in infant sera, indicating the transfer of maternal antibodies to the offspring [64,65]. It is likely that vaccination substantially reduces the incidence of viremia, thereby decreasing the probability of SP. Currently, the CDC recommends the COVID-19 vaccine for pregnant people.

## 8. Conclusions

Studies of the placentas of SARS-CoV-2-positive women revealed SP as a characteristic pattern associated with definite placental infection and chronic placental pathology, including maternal and fetal malperfusion, observed in the absence of placental involvement by the virus. SP is rare and frequently asymptomatic, but carries significant risks of stillbirth due to the massive destruction of the placental barrier and related fetal hypoxic damage. Factors predisposing to SP are not known; in a small set of data, maternal thrombophilia and pre-existing fetal growth restriction were statistically significant [38].

The data on increased rates of chronic lesions without associated placental infection in the placentas of women with COVID-19 are conflicting; these lesions are likely associated with underlying maternal comorbidities (diabetes, hypertension, obesity, and preeclampsia), which increasingly predispose patients to SARS-CoV-2 infection and become exacerbated in moderate to severe COVID-19.

The mechanisms underlying the rare predisposition to SP in late pregnancy remain unknown, and this area requires further investigation as COVID-19 becomes endemic. The elucidation of the molecular pathogenesis of SP could help develop screening tools to identify women at risk, which would improve the management of late pregnancies and potentially reduce the fetal mortality associated with this rare condition.

## Figures and Tables

**Figure 1 jdb-11-00042-f001:**
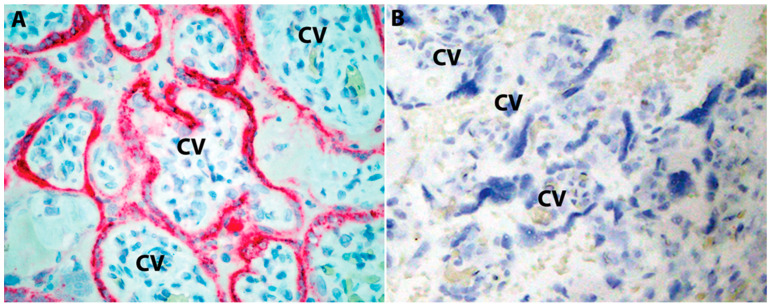
SARS-CoV-2 immunohistochemistry in infected placenta. (**A**) Strong granular staining (red chromogen) observed in the trophoblastic lining of chorionic villi (CV) in a case of SP in the third trimester. SARS-CoV-2 anti-spike mouse monoclonal antibody (mAb 1A9), dilution 1:1000; GeneTex, Irvine, CA, USA); (**B**) unaffected gestational-age-matched control. Original magnifications X400. Image from the author’s collection (the picture has not been previously published).

**Figure 2 jdb-11-00042-f002:**
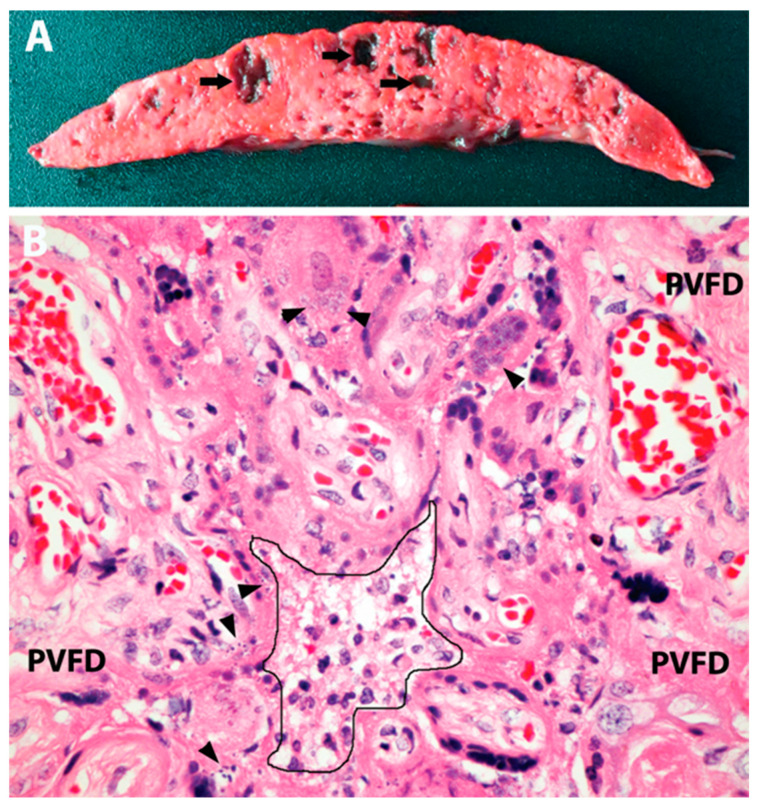
Placental pathology in 3rd-trimester stillbirth attributed to SARS-CoV-2 placentitis. (**A**) Gross picture of a slice of placental tissue demonstrating diffuse effacement and consolidation of the parenchyma with scattered acute hemorrhagic cavities (arrows). (**B**) Microscopy showing features of SP, including mild mixed inflammatory infiltrate in the intervillous space (center, outlined), trophoblast necrosis and apoptosis (arrowheads), and perivillous fibrin deposition (PVFD). Original magnification ×400. Image from the author’s collection (the picture has not been previously published).

**Table 1 jdb-11-00042-t001:** SARS-CoV-2 placentitis and clinicopathologic characteristics of associated stillbirth and early neonatal death. Summary of literature data.

Publication	Number of Cases	Extent of Placental Involvement by Destructive Pathology (PVFD)	Number of Autopsies Performed	Fetal Hypoxic–Ischemic Injury Identified	SARS-CoV-2 Detected in Fetal or Neonatal Tissues	Duration of Infection and/or Fetal Distress	Maternal SARS-CoV-2 Severity	Associated Conditions
Fitzgerald et al. [36]	6	80–90%	5	Not Reported	0	2–21 days	5 mild 1 asymptomatic	Thrombocytopenia (3 of 6)
Schwartz et al. [43]	68	35–100%, mean 77.7%	30	19	16	≈2 weeks	No severe COVID-19 reported	Not Reported
Konstantinidou et al. [38]	6	>75%	3	3	0	3–5 days	4 mild 2 asymptomatic	Thrombophilia FGR Multiple Sclerosis
Zaigham et al. [51]	7	>72%	4	4	2	2–25 days	4 mild-to-moderate 3 asymptomatic	No significant reported
Nielsen et al. [39]	10	60–90%	8	Not Reported	3	12 days on average	1 severe 8 mild-to-moderate 1 not reported	No significant reported

## Data Availability

Not applicable.

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
