# Peer review of "SARS-CoV-2 Infection in Late Pregnancy and Childbirth from the Perspective of Perinatal Pathology"

_jdb, 2023, doi:10.3390/jdb11040042_

Round 1

Reviewer 1 Report

Comments and Suggestions for Authors

Introduction: The introduction is very poor and must be improved. In particular: since author discussed the role of sars-cov-2 infection in a non respiratory disease (which is the main target of this infection), it deserves to be pointed out that sars-cov-2 infection can also lead to non respiratory complications (see PMID: 35943095, PMID: 35114008, PMID: 37649125). Since this is a review article, the multifaceted role of sars-cov-2 infection must be highlighted.

Line 44: A short introduction of PE pathology deserves to be added since it is just mentioned. In fact, the author takes it for granted that the reader knows what preeclampsia is. This pathology must be at least introduced highlighting its main characteristics such as hypertension and proteinuria. In addition, PE pregnancies are characterized by  inflammation, oxidative stress, endothelial dysfunction and trophoblast immaturity (see PMID: 33042016, PMID: 37296665).

Fig 1-2: These images are published or unpublished?

The author should add a table summarizing the results discussed in each study.

Author Response

Thank you very much for reviewing the manuscript.

  1. Introduction

Introduction has been re-written. Because this review is about placental and fetal pathology in SARS-CoV-2 infection, the introduction focuses on the effects of COVID-19 on feto-maternal health.

  1. Line 44

A new paragraph with definitions, pathogenesis, and risk factors of preeclampsia (with new corresponding references) has been added to the review.

  1. Fig 1-2: These images are published or unpublished?

The pictures have not been published and this is now stated in the footnotes.

  1. The author should add a table summarizing the results discussed in each study.

The summary table (Table 1) has been added.

All changes are highlighted in red.

Thank you for your review again,

LD

Reviewer 2 Report

Comments and Suggestions for Authors

This review reports on SARS-CoV-2 infection in placental and fetal tissues, focusing on the perspective of perinatal pathology. The manuscript is overall sound and well written.

For detailed comments, see the attachment.

Author Response

Thank you for reviewing the manuscript.

Comment 1 (vaccination): a paragraph on vaccination and SARS-CoV-2 placentitis has been added (the last paragraph of the “Biology of placental response”.

Comment 2 (predisposing factors): a paragraph on thrombophilia/thrombocytopenia associated with SARS-CoV-2 placentitis with massive perivillous fibrin deposition has been added to the chapter “biology of placental response (paragraph 4) as well as to the abstract.  The suggested reference has been also added. All changes are highlighted in red.

Thank you for your review again,

LD

Round 2

Reviewer 1 Report

Comments and Suggestions for Authors

Manuscript has significantly improved, accepted in present form.